# HierBatching: Locality-Aware Out-of-Core Training of Graph Neural Networks

## Abstract

Graph neural networks (GNNs) have become increasingly popular for analyzing data organized as massive graphs. Efficient training of GNN models under limited computing resources is critical for GNN's widespread adoption. We consider the use of a single commodity machine with limited memory (e.g., 128GB) but ample external storage (e.g., 1TB). On such a platform, the feature data or even the graph may not fit in the memory. When data is stored on external storage, gathering features and constructing neighborhood subgraphs in a typical mini-batch training incurs random storage accesses and thus, causes expensive data movement.

To overcome this bottleneck, we propose a *locality-aware* training scheme, coined HierBatching, which significantly increases training speed while retaining the training quality. The key idea is to exploit the memory hierarchy of a modern GPU machine by constructing batches in an analogously hierarchical manner. HierBatching groups nodes in partitions, each of which is laid out contiguously in the disk for maximal *spatial locality*. Meanwhile, the main memory is treated as a cache that holds a mega-batch, which is a random collection of partitions. Mini-batches are sampled for GPU training from the mega-batch in the main memory. Each mega-batch is reused multiple times to improve *temporal locality*. Our experiments show that on a machine with 128GB main memory, HierBatching is **3×** to **20×** faster than a straightforward out-of-core training approach by using mmap, while maintaining the prediction accuracy.

## 1 Introduction

Graph neural networks (GNNs) have emerged as effective machine learning models for many practical applications, such as social network analysis, financial forensics, recommendation, and traffic forecasting (Hamilton et al., 2017b). Training GNNs has become increasingly challenging as the size of the graphs has increased rapidly. Even benchmark datasets that mimic real-life applications have grown to sizes that forbid the use of commodity laptops or servers for experimentation and model development. For example, the `MAG240M` dataset (Hu et al., 2020) has 407GB of raw data (where 349GB accounts for node features). The storage requirement easily exceeds the memory capacity of a single machine; hence, one considers either exploiting external storage or using a distributed-memory cluster. The latter imposes a natural cost barrier for many individuals and organizations, and also has been demonstrated to be often heavily communication-bound (Ramezani et al., 2022).

In this work, we exploit external storage (e.g., SSD) and perform *out-of-core* GNN training on a single machine with one or a few GPUs. Fig. 1(a) illustrates the typical memory hierarchy of such a machine. The ample external storage is assumed to be able to store the entire dataset in a format that facilitates processing and training, while the main memory cannot. Each GPU, as is common in machine architectures encountered nowadays, has an even smaller memory capacity than the main memory. We use one or multiple GPUs to perform mini-batch training.

Traditional wisdom suggests that key to the exploitation of the memory hierarchy of a machine is *locality*: *spatial locality* that takes advantage of sequential data accesses and thus reads data in blocks instead of single words, and *temporal locality*, which takes advantage of the fact that recently accessed data will be accessed again in the near future, and thus, put it at a place closer to the processors to reduce access time. In disk-based computing, for spatial locality the operating system retrieves data from disk to memory in fixed-sized 4KB to 8KB blocks, called pages. This avoids the read

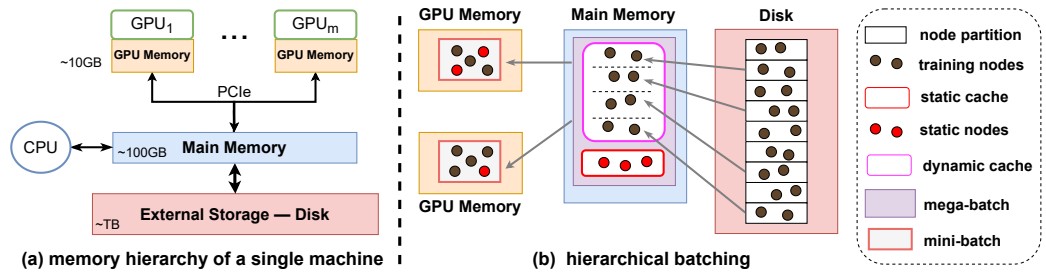

Figure 1: Overview of hierarchical batching. The disk stores the entire graph which is divided into partitions. Multiple (four in this example) partitions are randomly selected and loaded into the main memory to form a mega-batch, from which mini-batches are sampled and moved to the GPUs for training.

amplification problem (which dramatically reduces the effective disk bandwidth) when disk is read in cache-line-size blocks. For temporal locality, it is desirable that data loaded from the disk be reused as often as possible, given the significant latency caused by moving data from disk to memory.

Following this intuition, we propose a *hierarchical batching* scheme, abbreviated as HierBatching, to exploit locality in modern commodity memory hierarchy (external storage, main memory, and GPU memory). HierBatching batches data in an analogously hierarchical fashion: the entire set of graph nodes (with features and edge lists) is stored in external storage, while a *mega-batch* of nodes is sampled and copied to the main memory serving as a cache; mini-batches are then sampled from the cache for gradient-based training. We make this scheme *locality-aware* by partitioning the graph nodes into many small partitions that are stored consecutively in the disk (spatial locality), and performing mini-batch training with the same cache multiple rounds (temporal locality). HierBatching preserves the random nature of a stochastic training, as it forms each mega-batch with random combinations of partitions and samples mini-batches from the mega-batch randomly.

A realization of HierBatching incurs a subtle challenge because some of the neighbors of a node in a mega-batch may not be included in the cache, thereby degrading the training quality and the prediction accuracy. To maximally increase the node degrees with a small memory, we propose to permanently store the highest-degree nodes in the memory, because they are more likely to be connected to nodes in the mega-batch. We thus divide the main memory into a *static cache* and a *dynamic cache* and use the former to accommodate these nodes, such that their features are always reachable without repetitive transfers from the external storage. See Fig. 1(b) for a preview.

Our work makes the following contributions:

- We study a practical but under-explored scenario for training GNNs on massive graphs using external storage and propose HierBatching, a locality-aware batching scheme that fully leverages the memory hierarchy of a machine, particularly disk, to improve training efficiency.

- We introduce a static cache to compensate the loss of node degrees in the formation of the batching hierarchy, retaining model accuracy obtained by standard in-memory mini-batch training.

- We demonstrate empirically that, on a GPU equipped machine with 128GB main memory, Hier-Batching is **3×** to **20×** faster than DGL with `mmap` support, while retaining prediction accuracy. HierBatching is also competitive with in-memory DGL, which requires 3 times more memory.

## 2  PRELIMINARIES AND RELATED WORK

In this work, we consider message-passing GNNs that act on a given graph $G(V, E)$, where $V$ is the node set and $E$ is the edge set. For each node $v \in V$, let $\mathbf{h}_v^0$ be the initial feature vector. A $K$-layer GNN uses message passing to iteratively update the feature vector and produces an output vector $\mathbf{h}_v^K$. Specifically, the update at the $k$-th layer ($1 \leq k \leq K$) reads

$$\mathbf{h}_v^k = \texttt{update}^k\Big(\mathbf{h}_v^{k-1}, \texttt{aggregate}^k\big(\{\mathbf{h}_u^{k-1} \mid u \in \mathcal{N}(v)\}\big)\Big),$$

where $\mathcal{N}(v)$ denotes the 1-hop neighborhood of $v$, and $\texttt{update}^k$ and $\texttt{aggregate}^k$ are operators on the feature vectors, generally layer-dependent. Common GNNs, such as GCN (Kipf & Welling,

2016), GraphSAGE (Hamilton et al., 2017a), GAT (Veličković et al., 2018), and GIN (Xu et al., 2019), follow this framework but differ in the design of the two operators.

To scale GNNs to massive datasets, a frequently studied approach is distributed-memory training (Yang, 2019; Zhang et al., 2020; Zheng et al., 2020; Jia et al., 2020; Tripathy et al., 2020; Hoang et al., 2021; Md et al., 2021; Gandhi & Iyer, 2021; Kaler et al., 2022). However, such an approach often requires frequent and voluminous inter-machine communication, including features, edge lists, models, and gradients, causing substantial challenges in parallelization, synchronization, and pipelining. LLCG (Ramezani et al., 2022) reduces communication by using local training and global correction; however, it requires a powerful global server that can process the entire graph, at odds with a limited memory budget.

GNN training is generally done by following the mini-batch training scheme, propelled by the advances in stochastic optimizations and widely adopted by the designers of neural networks. Mini-batch training on graph data, however, suffers the well-known *neighborhood explosion* problem: for a $K$-layer GNN, the loss of a training example requires information of its $K$-hop neighborhood, whose size is exponential in $K$ in the worst case. Therefore, sampling techniques (Hamilton et al., 2017a; Chen et al., 2018; Chiang et al., 2019; Zeng et al., 2020) have been proposed to limit the size of the neighborhood. This *batching-plus-sampling* approach effectively reduces memory consumption, but data movement remains a bottleneck that hampers training efficiency. Prior work that aims at reducing data movement exists. GNS (Dong et al., 2021), PaGraph (Lin et al., 2020) and GNNLab (Yang et al., 2022) all employ GPU memory as a cache of main memory and keep likely-reusable data for fast accesses; GNS further prioritizes sampling from the cache nodes to reduce the cache misses. LazyGCN (Ramezani et al., 2020) recycles the already-sampled data in GPU memory, also reducing sampling and data movement overhead. However, none of the approaches handles the out-of-core scenario; in particular, they do not eliminate random accesses or consider the spatial locality, which are critical for data stored in the disk. MariusGNN and its prequel (Waleffe et al., 2022; Mohoney et al., 2021) are rare works that studied out-of-core training. Yet, they employ random partitioning that could render the in-memory graph overly sparsified and hurt the GNN prediction accuracy. A detailed analysis of MariusGNN and a comparison with our approach are provided in Appendix D.2.

Other approaches to scaling up GNNs include GNNAutoScale (Fey et al., 2021), graph coarsening (Huang et al., 2021), compression (Liu et al., 2021), and quantization (Ding et al., 2021). They are orthogonal to our approach and can be applied together with HierBatching in practice.

## 3   LOCALITY-AWARE HIERARCHICAL BATCHING

The typical mini-batching and sampling method for training GNNs works well in the in-memory training setting, but is not efficient for the out-of-core setting. This is because the batching approach is not aware of the memory hierarchy in a real machine. More specifically, external storage accesses are much slower than memory accesses. The random nature of mini-batching causes random disk accesses, rendering out-of-core training orders of magnitude slower than in-memory training. Even if disk data is treated as a memory-mapped file (`mmap`), page alignment will inevitably cause movement of unneeded data from disk to memory. Moreover, *neighborhood sampling* can be rather inefficient, if in-memory sampling is interleaved with moving edge lists from disk to memory, hop by hop.

We thus propose HierBatching, an out-of-core scheme that effectively leverages the memory hierarchy for efficient training. The development of HierBatching undergoes a series steps, including exploiting *spatial locality* through graph partitioning (§3.1), compensating degree loss by using a *static cache* (§3.2), and improving *temporal locality* through *mega-batch reuse and pipelining* (§3.3). We summarize the overall training algorithm in §3.4 and analyze the training speedup in §3.5.

### 3.1   HIERARCHICAL BATCHING AND SPATIAL LOCALITY

The first step of an efficient solution is to create a hierarchy for the batches, analogous to the memory hierarchy of a typical machine for training neural networks (external storage – main memory – GPU memory; see Fig. 1(a)). We treat the main memory as a *cache* for the disk and call data residing in the main memory a *mega-batch*. Once the mega-batch is formed through disk I/O, we may sample mini-batches from it, eliminating extra disk accesses. It has been demonstrated that if the mega-batch

is sampled randomly from the entire graph, training will converge and the resulting model is as good as that trained without using mega-batches (Ramezani et al., 2020).

The solution above reduces disk accesses, but the accesses are still random. To improve *spatial locality*, we divide the nodes into *partitions* and randomly sample partitions instead of nodes to form a mega-batch. Node data of a partition, i.e., feature vectors and edge lists, are laid out contiguously in disk, so that accesses to the nodes in the same group are sequential. An obvious benefit of grouping is the reduction of random disk accesses; moreover, it leads to the I/O of more consecutive data, reducing the collateral movement of unneeded data due to page alignment. This mechanism is illustrated in Fig. 1(b), where a box denotes a partition and the dots inside are the training nodes.

The number of partitions is chosen to balance two factors: too large a partition weakens the randomness of the mega-batch, compromising training quality; too small a partition weakens spatial locality, slowing down the disk data transfer. One rule of thumb is to ensure that the edge lists of a partition occupy only a few pages, given a specific page size and the average node degree. Moreover, it is desirable to use the minimum edge cut as the partitioning objective, so that the mega-batch, as a set of partitions, loses as few neighbors as possible. For a proof of concept, we use the off-the-shelf partitioner METIS (Karypis & Kumar, 1998) in our experiments. In cases when the graph storage is beyond the memory capacity, an out-of-core partitioner (Kaur & Gupta, 2021) is used.

## 3.2 Static and Dynamic Caches

Let the graph be divided into $N_p$ disjoined partitions and the capacity of the cache (main memory) allows holding $N_c$ ($< N_p$) partitions. In each training epoch, we randomly shuffle the partitions and move $N_c$ partitions from the disk to perform in-memory training. The cache contains a subgraph induced by the nodes inside these $N_c$ partitions. We enumerate mini-batches and sample neighbors for each layer from this subgraph to perform model updates. Compared with the full graph, however, such a subgraph will inevitably lose edges between the nodes in it and those outside it. This fact causes the degradation of prediction accuracy on some of the large graphs we experimented with.

To mitigate the performance loss due to insufficient neighbors, we permanently store in the memory some high-degree nodes that are helpful to increase the overall node degree. The part of the memory that holds these nodes is called the *static cache*. The rest of the memory acts as a *dynamic cache*, which holds the randomly selected partitions from the disk. As we can only allocate a small amount of memory for the static cache, we want to maximize its usage. A natural question is which static nodes are the most beneficial. Let $C_S$ with cardinality $k$ be the node set in the static cache and let $\{C_D^{(i)}\}$ be the node set in the dynamic cache for the $i$-th mega-batch. Then, the objective is to maximize the sum of increased degrees over all mega-batches:

$$\max_{C_S \subseteq V, \ |C_S|=k} \sum_i \Delta(C_D^{(i)}, C_S), \qquad (1)$$

where for each mega-batch,

$$\Delta(C_D, C_S) := \sum_{u \in C_D} \sum_{v \in C_D \cup C_S} \mathbb{1}(u, v) - \sum_{u \in C_D} \sum_{v \in C_D} \mathbb{1}(u, v), \qquad (2)$$

with $\mathbb{1}(u, v)$ being the indicator function which denotes the existence of an edge between nodes $u$ and $v$. It is not hard to see that the solution to Eq. (1) is

$$C_S^{\text{opt}} = \{ \text{ first } k \text{ nodes } v \text{ in the decreasing order of } \text{degree}(v) - \text{degree}_{\text{sub}}(v) \}, \qquad (3)$$

where $\text{degree}(v)$ and $\text{degree}_{\text{sub}}(v)$ denote the degree of $v$ in the full graph and the degree in the subgraph induced by $C_D^{(i)}$, respectively. See Appendix A for a proof.

Note that the optimal selection for the static cache derived in Eq. (3) is not "static": it depends on the content in the dynamic cache, as $\text{degree}_{\text{sub}}(v)$ changes over epochs due to the shuffling of partitions. In practice, we use $\text{degree}(v)$ to approximate $\text{degree}(v) - \text{degree}_{\text{sub}}(v)$ when forming the static cache; that is, it contains the $k$ nodes with the highest degrees. Inclusion of these nodes is intuitively sensible: a high-degree node connects more neighbors and offers a larger degree increase. Moreover, if all nodes have similar degrees in the subgraph as those in the full graph, then the first $k$ nodes in the ordering of $\text{degree}(v)$ would be similar to those in Eq. (3).

Fig. 2 illustrates the degree increase due to the use of a static cache, among nodes grouped by their degrees in the mega-batch. The orange part is the total degree increase for all nodes in the group. One sees that the group of lowest-degree nodes (0-5) incurs the most increase, benefiting the most significantly from the inclusion of static nodes; whereas the highest-degree nodes (>50) incurs the least amount of degree increase. Implementation details of the static cache are in Appendix C.1.

## 3.3 Pipelining and Mega-batch Reuse

Given a mega-batch, we enumerate mini-batches of training nodes from it to perform training on GPUs. We define a "pass" as using every training node in the mega-batch exactly *once*. The data transfer from disk to form a mega-batch usually takes more time than that of executing one pass on GPUs. This means HierBatching's speed can still be limited by the disk data transfer time. On the other hand, for datasets with a low training node ratio, performing one pass may *underutilize* the data in the memory, i.e., some of the nodes in the mega-batch may not even be sampled.

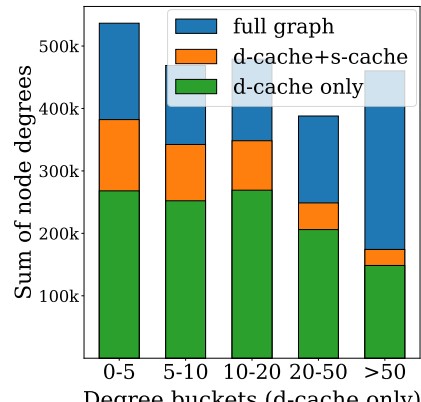

Hence, we employ two techniques cooperatively, *pipelining* and *mega-batch reuse*, to overcome this bottleneck. The idea is to repeatedly use the *current* mega-batch in memory while waiting for the disk transfer for the *next* mega-batch, in an exchange of lowering the number of epochs required to reach training convergence. In the context of HierBatching, *pipelining* refers to overlapping the computation, i.e., training with the current mega-

Figure 2: Node degree increase for five groups of nodes, bucketed by their degrees in the mega-batch. For each group, the green bar shows the sum of degrees before adding the static cache. The orange bar is the increase of degrees due to the inclusion of static cache, while the blue bar shows the loss of degrees.

batch, with communication, i.e., the data transfer of the next mega-batch. In addition, *mega-batch reuse* lets the training go through a mega-batch in multiple passes to exhaust the disk transfer time for the next mega-batch. Repeated use of the mega-batch in the cache improves *temporal locality*.

Training with a mega-batch in multiple passes deviates from the usual setting of stochastic optimization. In our experiments it always converges, but its theoretical analysis is rather challenging and is beyond the scope of this paper. When done in one pass, the training is similar to ClusterGCN (Chiang et al., 2019), whose empirical success in convergence supports our proposal. On the other hand, if we are willing to assume that the mega-batches and the mini-batches therein are all sampled with replacement, the setting falls back to standard stochastic gradient descent and we do obtain convergence guarantees (Bottou et al., 2018). The challenge of analysis lies in the fact that these batches are sampled without replacement. Only very recently, analysis of such a setting was conducted and even so, it considered only strongly convex objective functions (HaoChen & Sra, 2019). We leave the analysis as future work but supplement it through empirical evidences in §4.1.

## 3.4 Overall Algorithm

Let us summarize the training procedure in Algorithm 1. We assume that the graph has been partitioned and each of the $N_p$ partitions is laid out contiguously on external storage. First, fill the static cache (line 9). Then, we train in epochs (line 10), each of which performs a sweep over the entire training set. In each sub-epoch (line 11), $N_d$ partitions are sampled and loaded to main memory, forming a maga-batch (lines 12–15). This part includes loading node features and edge lists and forming the subgraph induced by the nodes in the dynamic cache and the static cache. Furthermore, this part is pipelined with the rest of the algorithm, which performs GNN training in memory. Therein, the mega-batch is recycled in $p$ passes (line 17). For each pass, a standard multi-GPU training procedure is followed to update the model parameters (lines 18–23).

---

**Algorithm 1** Out-of-core GNN training with hierarchical batching (multi-GPU single-machine)

---

1: **Input**
2: $\quad G(V, E)\quad$ input graph, pre-partitioned, with nodes in the same partition stored contiguously in disk
3: $\quad N_p\quad$ total number of partitions
4: $\quad N_d\quad$ number of partitions in the dynamic cache
5: $\quad r_s\quad$ the ratio of nodes in $V$ pinned in the static cache
6: $\quad p\quad$ number of passes per mega-batch
7: $\quad B\quad$ mini-batch size, i.e., number of training nodes in a mini-batch
8: $\quad q\quad$ number of GPUs

9: Let the set of $r_s \times |V|$ highest-degree nodes be $\mathcal{P}$. Move features of $\mathcal{P}$ to the main memory. $\triangleright$ static cache
10: **for each** epoch **do**
11: $\quad$ **for each** $i \in [0, \frac{N_p}{N_d})$ **do** $\qquad\qquad\qquad\qquad\qquad$ $\triangleright$ hierarchical batching at the main memory level
$\qquad\qquad\qquad\qquad\qquad\qquad\qquad\qquad\qquad$ $\triangleright$ lines 12–15 are pipelined with lines 16–23

12: $\qquad$ Random sample $N_d$ node partitions and let the sampled node set be $\mathcal{T}$
13: $\qquad$ Move the features of $\mathcal{T}$ to the dynamic cache in the main memory $\qquad$ $\triangleright$ disk data transfer
14: $\qquad$ Move the edge lists of $\mathcal{T} \cup (\mathcal{N}(\mathcal{T}) \cap \mathcal{P})$ to the main memory $\qquad$ $\triangleright \mathcal{N}(\mathcal{T})$ is $\mathcal{T}$'s neighborhood
15: $\qquad$ Form a subgraph $\mathcal{G}_{cache}$ with node set $\mathcal{T} \cup (\mathcal{N}(\mathcal{T}) \cap \mathcal{P})$ $\qquad$ $\triangleright$ a new mega-batch formed

16: $\qquad$ Let $S_{\mathcal{T}}$ be the number of training nodes in $\mathcal{T}$, and $r = \frac{S_{\mathcal{T}}}{B}$ be the number of mini-batches per pass
17: $\qquad$ **for each** pass $j \in [1, p]$ **do** $\qquad\qquad\qquad\qquad$ $\triangleright$ reuse the mega-batch in the main memory
18: $\qquad\quad$ Shuffle training nodes in $\mathcal{G}_{cache}$ into GPU-level mini-batches $\{\mathcal{B}_1, \ldots, \mathcal{B}_{q \times r}\}$
19: $\qquad\quad$ **for each** mini-batch $k \in [0, r)$ **do** $\qquad\qquad$ $\triangleright$ hierarchical batching at the GPU level
20: $\qquad\qquad$ **for each** GPU $d \in [1, q]$ **in parallel do** $\qquad\qquad$ $\triangleright$ multi-GPU training
21: $\qquad\qquad\quad$ Sample a message-flow graph $\mathcal{G}_s$ from $\mathcal{G}_{cache}$ for the mini-batch $\mathcal{B}_{k \times q + d}$
22: $\qquad\qquad\quad$ Move $\mathcal{G}_s$ to the GPU and compute the gradient
23: $\qquad\qquad$ All GPUs average the gradient and update the model

---

## 3.5 SPEEDUP ANALYSIS

We analyze the speedup of the proposed locality-aware hierarchical batching scheme, over a straightforward out-of-core training by using `mmap` without exploiting locality. We use the same notation in Algorithm 1, in which we run $p$ passes and execute $p \times r$ mini-batches for each mega-batch. Let $D$ be the node feature dimension and $S_{feat}$ be the data precision in bytes. Let $BW_{seq}$ and $BW_{rand}$ be the disk bandwidth when performing sequential accesses and random accesses, respectively. Generally, $BW_{rand}/BW_{seq} := \alpha \approx S_{acc}/S_{opt}$, where $S_{acc} = D \times S_{feat}$ is the size of a single random access and $S_{opt}$ is the optimal disk request size that leads to sequential access. For example, in the MAG240M dataset, $D = 768$ and $S_{feat} = 2B$, while $S_{opt} = 4KB$ for a modern flash disk. Then, $\alpha = 768 \times 2/4069 = 0.375$, which means that random access of a node feature vector leverages only 37.5% of the peak disk bandwidth.

For simplicity of analysis, we neglect the disk transfer of the graph, as loading node features is way more time consuming for large $D$. Using `mmap` without locality, the time to load one mini-batch is $t_{disk}^{mini} = B \times F \times D \times S_{feat}/BW_{rand}$, where $B$ is the mini-batch size and $F$ is the sampling fanout. Let $\gamma$ be the ratio of training nodes in the graph, and we assume even distribution of training nodes across mega-batches, so that $S_{\mathcal{T}}/C_d \approx \gamma$. Let the time to train one mini-batch be $t_{gpu}$, which includes data transfer from memory to GPUs, doing forward and backward calculations, averaging gradients, and updating the model. Typically, $t_{disk}^{mini}/t_{gpu} := \beta > 1$. Assuming pipelining, the majority time to train $p \times r$ mini-batches is $t_{mmap} = p \times r \times t_{disk}^{mini}$. On the other hand, for HierBatching, loading one mega-batch takes time $t_{disk}^{mega} = C_d \times D \times S_{feat}/BW_{seq}$ and thus training takes time $t_{hier} = \max\{t_{disk}^{mega}, p \times r \times t_{gpu}\}$. Therefore, the speedup is $t_{mmap}/t_{hier} = p \times \gamma \times F/\alpha$ if disk access takes more time in HierBatching; otherwise, $t_{mmap}/t_{hier} = \beta$.

## 4 EVALUATION

**Datasets and evaluation metric.** We use large benchmark datasets listed in Table 1. Mag240M and ogbn-* datasets come from the Open Graph Benchmark Hu et al. (2020; 2021). Mag240M-C is the subgraph extracted from Mag240M that contains only paper citation edges. All graphs are made undirected if they are originally not. The data statistics after the transformation are listed in Table 1. On each dataset, the task is to predict node labels. Although smaller datasets such as ogbn-arxiv and ogbn-products could easily fit into the main memory in our setting, they are included in the evaluation to demonstrate the competitive model quality produced by HierBatching.

Table 1: Summary of evaluated datasets

| Dataset | # Nodes | # Edges | Feature | Classes | Multi-label | Train / Val / Test | Raw data size |
|---|---|---|---|---|---|---|---|
| ogbn-arxiv | 169,343 | 2,332,486 | 128 | 40 | No | 0.54 / 0.18 / 0.29 | 123MB |
| ogbn-products | 2,449,029 | 123,718,024 | 100 | 47 | No | 0.08 / 0.02 / 0.90 | 2.8GB |
| ogbn-papers100M | 111,059,956 | 3,231,371,744 | 128 | 172 | No | 0.01 / 0.001 / 0.002 | **103GB** |
| Mag240M-C | 121,751,666 | 2,595,497,852 | 768 | 153 | No | 0.01 / 0.001 / 0.001 | **214GB** |
| Mag240M | 244,160,499 | 3,456,728,464 | 768 | 153 | No | 4.5e-3 / 6e-4 / 6e-4 | **407GB** |

**Evaluation platform.** We use two compute platforms in our experiments, a local cluster and AWS. Each compute node in the local cluster has 32 CPU cores, 1TB DRAM and 4 NVIDIA V100 GPUs. The AWS instance has 24 CPU cores (2-way hyperthreading, 48 threads), 386GB DRAM and 4 NVIDIA T4 GPUs. The AWS machine is equipped with a fast local flash storage while the local cluster stores the datasets in a networked file system. The difference of the two is useful in showing how I/O throughputs could affect training speed. We regulate the DRAM size on both platforms, e.g., from 128GB to 386GB, to meet the memory budget assumptions of different experiments.

**Experimental Setup.** We implement HierBatching based on PyTorch Paszke et al. (2019) and DGL Wang et al. (2019). We use the *micro F1-score* to measure accuracy. For accuracy-related experiments, we run the training for a fixed number of epochs as listed in Table 4 in the Appendix. We report the test scores produced by the model at the best validation epoch during the training. To obtain the training time per epoch, we run each experiment for five epochs and take an average. Further details of our experimental settings are deferred to Appendix C.

### 4.1 ACCURACY COMPARISON WITH STANDARD NEIGHBORHOOD SAMPLING

To demonstrate that HierBatching retains the accuracy of standard in-memory training schemes, we compare test accuracy of HierBatching (`HB`) with that of *neighbor sampling* (`NS`), the standard mini-batching and sampling method used to train large-scale GNNs (Hamilton et al., 2017a). `NS` assumes full graph accesses during sampling. We allocate as much memory as `NS` needs to run in memory. For `HB`, we choose static cache capacity $C_s = |V|/100$ and number of passes $p = 2$. For a given dataset and model, we keep common hyper-parameters the same (e.g. GNN layers, hidden dimensions, learning rates) across all methods. Detailed hyper-parameter settings are in Appendix C.

Table 2 lists the test F1-micro scores obtained by `NS` and `HB`. We observe that their resulting accuracies are nearly the same across models and datasets. However, without the static cache, `HB(-s)` suffers notable accuracy degradation ($\geq 1\%$) in many cases, as shown in red. This observation demonstrates the importance of static cache for HierBatching. We also observe that `HB` and the no mega-batch reuse version, `HB(-r)`, achieve a similar accuracy, which means that mega-batch reuse does not compromise the training quality. Note that similarly to `HB(-s)`, MariusGNN (Waleffe et al., 2022) also suffers accuracy degradation, as it does random partitioning without compensating the edge loss. We conduct a detailed comparison with MariusGNN in Appendix D.2.

### 4.2 CONVERGENCE RATES AND TRAINING TIME

Fig. 3 shows the model convergence in terms of epochs, for both `NS` and `HB` approaches. We observe that without the static cache, `HB(-s)` not only lowers final accuracy but converges at a much slower rate, while `HB(-r)` converges almost as fast as `NS`. Even though we follow a non-standard stochastic training approach in `HB` by reusing each mega-batches, the model still converges well and in some cases even faster than `NS` (e.g., ogbn-products), primarily because it perform twice as many gradient updates per epoch. More gradient updates do not necessarily translate into longer training time in `HB` since the system is bottlenecked by I/O, not gradient computation. As we will show immediately, the training time per epoch in `HB` is very much comparable to the in-memory `NS` approach.

Table 2: Comparison of test accuracies (F1-micro score) between HierBatching (`HB`) and neighbor sampling (`NS`). `HB` uses $C_s = |V|/100$ and $p = 2$. `HB(-s)` disables the static cache on top of `HB`, i.e., $C_s = 0$. `HB(-r)` disables mega-batch reuse on top of `HB`, i.e., $p = 1$. Significant accuracy drops are hignlighted in red.

| GNN Model | Dataset | NS | HB(-s) | HB(-r) | HB |
|---|---|---|---|---|---|
| **SAGE** | ogbn-arxiv | 71.52 | 70.99 | 71.43 | 71.42 |
| | ogbn-products | 78.90 | 78.98 | 78.90 | 78.84 |
| | ogbn-papers100M | 64.94 | 63.85 | 64.56 | 64.83 |
| | Mag240M-C | 65.92 | 64.94 | 65.69 | 65.98 |
| | Mag240M | 68.83 | 67.08 | 68.25 | 68.56 |
| **GAT** | ogbn-arxiv | 71.76 | 69.58 | 71.21 | 71.73 |
| | ogbn-products | 79.54 | 79.61 | 79.58 | 79.55 |
| | ogbn-papers100M | 64.58 | 61.45 | 64.31 | 64.20 |
| | Mag240M-C | 65.74 | 65.06 | 65.69 | 66.09 |
| | Mag240M | 68.40 | 67.33 | 68.07 | 68.89 |
| **GIN** | ogbn-arxiv | 70.24 | 68.61 | 70.22 | 70.10 |
| | ogbn-products | 75.70 | 76.58 | 76.15 | 76.77 |
| | ogbn-papers100M | 65.07 | 62.33 | 64.63 | 64.81 |
| | Mag240M-C | 63.73 | 62.93 | 63.59 | 63.36 |
| | Mag240M | 67.47 | 66.33 | 67.02 | 66.58 |

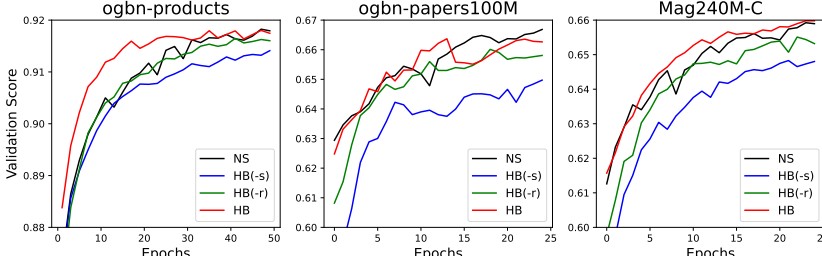

Figure 3: GraphSAGE convergence history for neighbor sampling and HierBatching.

We compare the training speed of HierBatching with two settings of DGL, DGL* and DGL+`mmap`. DGL* is the vanilla DGL which requires all data to be resident in the memory. It uses 386GB main memory. DGL+`mmap` adds the `mmap` [1] support on top of DGL to handle the case of insufficient main memory. This setting uses 128GB main memory.

Table 3 lists the training time per epoch on the AWS machine. The first observation is that HierBatching (`HB`) can run all of the cases, thanks to HierBatching's awareness of the memory hierarchy. In contrast, DGL runs out of memory for the largest dataset, Mag240M, which requires 349GB space to accommodate the feature data and 135GB for the graph structures (DGL maintains all three COO, CSC and CSR formats in memory). Even with `mmap` support, DGL+`mmap` can not run Mag240M within the limited memory budget, since it fails to hold the graph structures in memory.

The second observation is, `HB` runs significantly faster than `mmap`, and is only slightly slower than in-memory training. Compared with DGL+`mmap`, HierBatching achieves 2.6× to 20.4× speedup across different datasets and models (4.3× on average), mainly because of the improved spatial locality. As shown in §3.5, speedup depends on training nodes ratio $\gamma$ and fanout $F$, thus speedups vary across different datasets. Compared to DGL*, The training HierBatching can achieve competitive training speed with a maximum slowdown of 41% (GIN on mag240M-C), even though HierBatching has only one third of the memory capacity. Surprisingly, for some cases, e.g., GIN on ogbn-papers100M, `HB` is faster than DGL*. It turns out that these cases are GPU compute bound, and `HB` usually has smaller sampled graphs than DGL*, which means less GPU computation. Overall, these results demonstrate that HierBatching is extremely efficient in taking advantage of the system resources.

Notice that the AWS instance has a faster storage than the local cluster. We show the training time per epoch on the the local cluster in Table 6 in Appendix D. Due to slower network-based storage

---

[1] We use the PyTorch API `torch.Storage.from_file` to create `mmap`-ed tensors. It is infeasible to `mmap` graph structure data without significant modifications to the DGL codebase. Thus, we only `mmap` node features.

Table 3: Comparison of training time (sec) per epoch between HierBatching (HB), DGL* and DGL+`mmap`, on the **AWS** machine. DGL* uses 386GB memory, while HB and DGL+`mmap` use 128GB memory.

| Dataset | SAGE | | | GAT | | | GIN | | |
|---|---|---|---|---|---|---|---|---|---|
| | DGL* | DGL+`mmap` | HB | DGL* | DGL+`mmap` | HB | DGL* | DGL+`mmap` | HB |
| **ogbn-papers100M** | 202.6 | 1017.8 | 278.0 | 382.3 | 1426.3 | 385.8 | 822.2 | 9131.6 | 448.0 |
| **Mag240M-C** | 178.6 | 678.3 | 249.4 | 379.9 | 1042.7 | 326.9 | 183.5 | 664.6 | 258.3 |
| **Mag240M** | OoM | OoM | 521.3 | OoM | OoM | 546.6 | OoM | OoM | 528.3 |

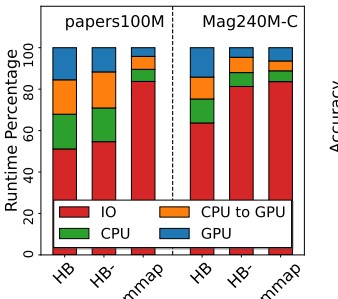 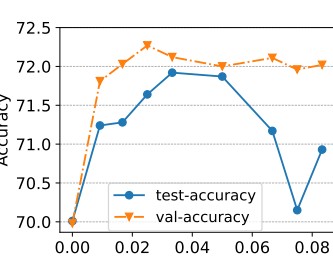 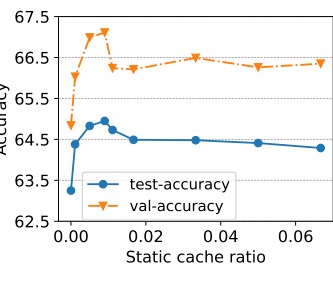

(a) ogbn-arxiv        (b) ogbn-papers100M

Figure 4: Training time break-down. Pipelining is disabled to better illustrate the overhead of each components.

Figure 5: Trade-off on static and dynamic cache sizes. 'Static cache ratio' refers to $C_s/|V|$. The overall budget for dynamic cache and static cache is $|V| \times 1\%$.

system, we observe even larger speedup (up to $58\times$) of HierBatching over DGL+`mmap`. Note that this network-based storage architecture is quite common in modern datacenters and supercomputing centers, which means HierBatching will be quite useful in the real systems.

## 4.3 ABLATION STUDY AND SENSITIVITY STUDY

In order to study the influence of spatial locality, we disrupt `HB`'s feature contiguous layout in disk, denoted as `HB-`, which thus access node features randomly and can not enjoy spatial locality. `HB-` differs from DGL+`mmap` as it forms mega-batches in memory. Fig. 4 reports the training time breakdown of three approaches. Interestingly, `HB-` exhibits different behaviors with two datasets. For ogbn-papers100M where the node feature size is relatively small, `HB-` allows the operating system to cache most of the feature data in the memory as training proceeds and yields almost no slowdown. This is not possible with DGL+`mmap` since the full graph structure takes almost all the available memory space, leaving little space for caching. In contrast, the node feature size of Mag240M-C far exceeds the memory budget and cannot be cached efficiently. In this case, `HB-` behaves very similarly with DGL+`mmap`, spending most of the time on gathering node features from the external storage.

Given a certain memory budget, the static cache capacity $C_s$ and dynamic cache capacity $C_d$ are two competing parameters. Increasing $C_s$ can keep more high-degree nodes in the memory, but also decreases $C_d$ and thus reduces randomness. Oppositely, increasing $C_d$ increases randomness but reduces the degrees compensation by the static cache. We conduct an experiment with a fixed memory quota for dynamic cache and static cache, and study how the accuracy changes as the static cache ratio changes in Fig. 5. As shown, all curves in both figures peak at a static cache ratio between $(0, 0.04)$, which demonstrates that there does exist a sweet spot to pick the sizes of the two caches.

## 5 CONCLUSIONS

We have proposed a novel locality-aware GNN training approach, HierBatching, for scaling up GNNs to massive graphs. HierBatching targets a single machine with multiple GPUs but a limited memory budget. The core of our approach is hierarchical batching, which hierarchically divides the training data to match the memory hierarchy of the machine. The main memory serves as a cache for the disk, and is fully utilized as a dynamic cache and a static cache, to balance training speed and prediction quality. Overall, our approach effectively improves both spatial and temporal locality. Experiment results demonstrate that HierBatching dramatically improves GNN training speed for massive graphs on a single machine, while retaining the prediction accuracy.

REPRODUCIBILITY STATEMENT

We provide a link to an ananymous Github repository hosting the code, scripts and instructions to reproduce results in our evaluation section: https://github.com/HierBatching/HierBatching. Detailed instructions on how to prepare the datasets, build and run the code are included in the README.txt file under the repository.

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

# A PROOF OF EQ. (3)

Simplifying Eq. (2), we obtain

$$\Delta(C_D, C_S) = \sum_{u \in C_D} \sum_{v \in C_S \setminus C_D} \mathbb{1}(u, v)$$

$$= \sum_{u \in C_D} \left( \sum_{v \in C_S} \mathbb{1}(u, v) - \sum_{v \in C_D \cap C_S} \mathbb{1}(u, v) \right) \quad (4)$$

$$= \sum_{v \in C_S} \sum_{u \in C_D} \mathbb{1}(u, v) - \sum_{v \in C_D \cap C_S} \sum_{u \in C_D} \mathbb{1}(u, v).$$

Define the partition assignment function $f$: $f(v) = i \iff v \in C_D^{(i)}$. Then, by substituting $\Delta(C_D^{(i)}, C_S)$ with Eq. (4) in Eq. (1), it follows that

$$\sum_i \Delta(C_D^{(i)}, C_S) = \sum_i \left( \sum_{v \in C_S} \sum_{u \in C_D^{(i)}} \mathbb{1}(u, v) - \sum_{v \in C_D^{(i)} \cap C_S} \sum_{u \in C_D^{(i)}} \mathbb{1}(u, v) \right)$$

$$= \sum_{v \in C_S} \sum_i \sum_{u \in C_D^{(i)}} \mathbb{1}(u, v) - \sum_i \sum_{v \in C_D^{(i)} \cap C_S} \sum_{u \in C_D^{(f(v))}} \mathbb{1}(u, v) \quad (5)$$

$$= \sum_{v \in C_S} \sum_{u \in V} \mathbb{1}(u, v) - \sum_{v \in C_S} \sum_{u \in C_D^{(f(v))}} \mathbb{1}(u, v)$$

$$= \sum_{v \in C_S} \left( \text{degree}(v) - \text{degree}_{\text{sub}}(v) \right).$$

Therefore, to maximize Eq. (5) with a cache budget $k$, we sort all nodes in $V$ according to $\text{degree}(v) - \text{degree}_{\text{sub}}(v)$ and take the first $k$ nodes.

# B CHOICE OF PARAMETERS IN ALGORITHM 1

*Choosing $N_p$.* Assume that $S_p = \frac{|V|}{N_p}$ is the size of a single partition, $D$ is the node feature dimension, $S_{feat}$ is the data precision in bytes, and $S_{opt}$ is the optimal disk request size that leads to sequential access. We choose $S_p$ to be no less than $S_{opt}/(D \times S_{feat})$, such that disk accesses are sequential. Hence, $N_p$ is no greater than $|V| \times D \times S_{feat}/S_{opt}$. As mentioned in §3.1, the partition should not be too large, either. In practice, we recommend choosing $N_p$ so that each partition consists of several thousands of nodes.

*Choosing $N_d$ and $r_s$.* Let $r_d = N_d/N_p$. The static cache capacity $r_s \times |V|$ and the dynamic cache capacity $r_d \times |V|$ are faced with a trade off: if the dataset has $A$ GB feature data and $B$ GB edge list data, we require $2(r_s + r_d) \times (A + B) \leq C$ where $C$ is the allocated memory space for the static and dynamic cache. The factor 2 comes from double buffering required for pipelining. Typical values of $r_s$ and $r_d$ in our experiments are $r_s = 0.01$ and $r_d \in [0.0625, 0.25]$.

*Choosing $p$.* It is straightforward to select the reuse ratio $p$ after training for a few epochs. Assume the time to load and construct and mega-batch is $T_{load}$, and the time to train a mega-batch with no reuse is $T_{comp}$. Since the two stages are pipelined, the latency between two mega-batches is $max\{T_{load}, T_{comp}\}$. Only when $T_{load} > T_{comp}$, we select a reuse ratio of $1 < p \leq T_{load}/T_{comp}$ to hide the remaining data-loading latency and guarantee that $T_{comp}$ never becomes the bottleneck.

Other parameters including the mini-batch size $B$ are common hyper-parameters present in most GNN training systems. We do not have additional constraints for those parameters.

## C  EVALUATION DETAILS

### C.1  IMPLEMENTATION DETAILS

Since static nodes are high-degree nodes, their edge lists may be too large to stay in the main memory, especially for power-law graphs. As a trade-off, only the features of the static nodes are stored in the cache, while the edge lists are kept in the disk. Because the static nodes are scattered in the disk, we store another copy of the edge lists for them in the disk contiguously, so that disk accesses remain sequential when edge lists are moved to the memory at every reloading of the dynamic cache (i.e., forming a new mega-batch). While loading edge lists, we only keep the edges of the induced subgraph of the mega-batch in memory, which needs much less memory space.

### C.2  CLARIFICATION OF DATASETS

The complete Mag240M dataset is a heterogeneous academic graph on which a relational GNN model would work the best. For ease of implementation, in the evaluation we treat it as a homogeneous graph and apply non-relational GNN models instead. Only minor accuracy drops are observed when compared with relational GNN baselines posted on OGB-LSC benchmark Hu et al. (2021). For datasets without publicly available test sets (Mag240M-C, Mag240M), we report the highest validation F1-scores. Considering the scales of large datasets, test and validation are performed with minibatching and sampling, similar to the practice in Kaler et al. (2022).

### C.3  HYPER-PARAMETER SETTINGS

Table 4 lists the key hyper-parameters we use in our experiment. Note that when evaluating the training speed, we use the same hyper-parameters, except that GAT uses hidden size of 512 in Table 3, but 1024 in Table 2 due to the smaller GPU memory on the AWS instance.

Table 4: Training configurations for results in Table 2

| Dataset | SAGE | | | | | |
| | Epochs | Hidden | Learning rate | Dropout | Fanouts | Layers |
|---|---|---|---|---|---|---|
| ogbn-arxiv | 100 | 256 | 0.001 | 0.5 | (15,10,5) / (50,50,50) | 3 |
| ogbn-products | 50 | 256 | 0.001 | 0.5 | (15,10,5) / (20,20,20) | 3 |
| ogbn-papers100M | 25 | 256 | 0.001 | 0.5 | (15,10,5) / (20,20,20) | 3 |
| Mag240M-C | 25 | 1024 | 0.001 | 0.5 | (25,15) / (25,15) | 2 |
| Mag240M | 25 | 1024 | 0.001 | 0.5 | (25,15) / (25,15) | 2 |
| Dataset | GAT | | | | | |
| | Epochs | Hidden | Learning rate | Dropout | Fanouts | |
| ogbn-arxiv | 100 | 256 | 0.001 | 0.5 | (15, 10, 5) / (50,50,50) | 3 |
| ogbn-products | 50 | 512 | 0.001 | 0.5 | (15, 10, 5) / (20,20,20) | 3 |
| ogbn-papers100M | 25 | 1024 | 0.001 | 0.5 | (15, 10, 5) / (20,20,20) | 3 |
| Mag240M-C | 25 | 1024 | 0.001 | 0.5 | (25,15) / (25,15) | 2 |
| Mag240M | 25 | 1024 | 0.001 | 0.5 | (25,15) / (25,15) | 2 |
| Dataset | GIN | | | | | |
| | Epochs | Hidden | Learning rate | Dropout | Fanouts | |
| ogbn-arxiv | 100 | 128 | 0.001 | 0.5 | (20,20,20) / (20,20,20) | 3 |
| ogbn-products | 50 | 512 | 0.003 | 0.5 | (15,10,5) / (20,20,20) | 3 |
| ogbn-papers100M | 25 | 512 | 0.003 | 0.5 | (20,20,20) / (20,20,20) | 3 |
| Mag240M-C | 25 | 1024 | 0.001 | 0.5 | (25,15) / (25,15) | 2 |
| Mag240M | 25 | 1024 | 0.001 | 0.5 | (25,15) / (25,15) | 2 |

Table 5 shows some extra parameters for HierBatching. $N_p$ is the total number of partitions. $N_d$ is the maximum number of partitions held in the dynamic cache. $r_s = C_s/|V|$ is the ratio of nodes kept in the static cache. $p$ is the number of passes per mega-batch.

Table 5: Extra training configurations for HierBatching in Table 2

| Dataset | $N_p$ | $N_d$ | $r_s$ | $p$ |
|---|---|---|---|---|
| ogbn-arxiv | 1024 | 256 | 0.01 | 2 |
| ogbn-products | 4096 | 512 | 0.01 | 2 |
| ogbn-papers100M | 16384 | 1024 | 0.01 | 2 |
| Mag240M-C | 16384 | 1024 | 0.01 | 2 |
| Mag240M | 16384 | 1024 | 0.01 | 2 |

## D  ADDITIONAL EXPERIMENTS

### D.1  TRAINING PERFORMANCE WITH THE LOCAL CLUSTER

Table 6 shows the training time per epoch on the local cluster. As it has faster GPUs (V100), its in-memory training, i.e., DGL*, is faster than that on the AWS machine. However, since it uses a network-based storage system, its random access latency is significantly larger than that of the AWS machine. Consequently, the DGL+mmap setting performs dramatically worse than that on AWS. In comparison, HierBatching (HB) becomes faster than that on AWS, as the storage system in the local cluster has similar sequential IO throughput as that of AWS, and HB can fully enjoy fast sequential accesses, thanks to the spatial locality aware design.

Table 6: Comparison of training time (s) per epoch between HierBatching (HB), DGL* and DGL+mmap on the **local cluster**. DGL* uses 386GB memory, while HB and DGL+mmap use 128GB memory. TO: timeout (>5 hours). ⋆ The unusual long training time with the Mag240M dataset is possibly due to the inefficiency of the network-based storage system when handling huge files. We forward the readers to Table 3 for a more realistic runtime ratios between Mag240M and other datasets.

| Dataset | SAGE | | | GAT | | | GIN | | |
|---|---|---|---|---|---|---|---|---|---|
| | DGL* | DGL+mmap | HB | DGL* | DGL+mmap | HB | DGL* | DGL+mmap | HB |
| **ogbn-papers100M** | 105.4 | 11037.5 | 190.3 | 254.2 | 11850.3 | 260.7 | 348.7 | TO | 238.8 |
| **Mag240M-C** | 77.7 | TO | 229.1 | 142.3 | TO | 185.8 | 75.7 | TO | 193.7 |
| **Mag240M⋆** | OoM | OoM | 1708.3 | OoM | OoM | 1828.5 | OoM | OoM | 1788.9 |

### D.2  COMPARISON BETWEEN HIERBATCHING AND MARIUSGNN

Both MariusGNN and HierBatching operate over pre-partitioned graph datasets, loading parts of the graph in the memory for training at one time. MariusGNN proposes two modes for training node-prediction GNNs:

- Sequential mode. When the training nodes and its induced subgraph could fit into the main memory, MariusGNN pins all training nodes and selects partitions randomly to form a partial in-memory graph. The graph remains unchanged throughout an epoch.

- Dispersed mode. When the sequential mode is not applicable, MariusGNN assumes the training nodes are dispersed across partitions. It then loads partitions into the in-memory buffer in a semi-random fashion until all training nodes appear once in the memory.

Our approach is similar to the dispersed mode of MariusGNN, with several key differences: 1) we identify the importance of min-cut graph partitioning while MariusGNN simply uses random partitioning; 2) MariusGNN in the dispersed mode does not keep a static cache; 3) MariusGNN never uses mega-batch reuse to hide extra IO latency.

An important component in MariusGNN is COMET, a fine-grained partition replacement policy. In addition to the number of partitions $p$, number of in-memory partitions $c$, it specifies one extra tunable parameter $l \leq c$ corresponding to the number of partitions to be replaced at each step. The partition sampling procedure in HierBatching could be seen as implementing a specific COMET policy with $l = c$. We argue that it is not a limitation of HierBatching. The value of $l$ actually has no influence on the overall IO traffic in the dispersed mode, since all partitions are loaded once in an epoch. Instead, a smaller $l$ would lead to less data randomness in the mega-batch and could affect the model accuracy. Although MariusGNN does not have analysis of how to choose $l$ for node prediction tasks, as a side note, MariusGNN suggests to make $l$ as large as possible for link prediction tasks (Section 4).

Compared to our work, the main concern with MariusGNN is that it employs no techniques to make up for the lost neighbors during the mega-batch construction. This has not been an big issue because the datasets used in MariusGNN are "easier" in a sense: e.g. for ogbn-papers100M if we only keep edges connected by training nodes for training and discard the remaining **99%** edges, the model could still achieve a test accuracy of 62.95%, an accuracy drop of merely 2% from training with the full graph. To reveal the risk of accuracy degradation in MariusGNN, we construct a new dataset by changing the data splits of ogbn-arxiv: 10% of nodes are chosen randomly as the training nodes and the rest as validation nodes. We compare the validation F1 scores of different approaches (we use the same set of hyper-parameters as in our paper):

- NS (training with full graph): 69.61±.07
- HB ($N_d/N_p = 1/16, r_s = 0.01$): 69.64±.08
- MariusGNN (sequential mode, $c/p = 1/8$): 67.08±.14
- MariusGNN (dispersed mode, $c/p = 1/8$): 67.29±.10

With this "harder" dataset, MariusGNN suffers from more significant accuracy losses, while HB still retains the model accuracy with NS. Therefore, we believe that HierBatching is a more robust out-of-core training solution than MariusGNN.

