# OpenReview forum: "HierBatching: Locality-Aware Out-of-Core Training of Graph Neural Networks"
_ICLR.cc/2023/Conference — Submitted to ICLR 2023_

### Official Review · Reviewer_p8K8 · 2022-10-20

**Confidence:** 4
**Correctness:** 3
**Technical Novelty And Significance:** 2
**Empirical Novelty And Significance:** 2
**Recommendation:** 3

**Clarity, Quality, Novelty And Reproducibility:**

The exposition is clear, but unnecessarily wordy and overly formal (e.g., Alg 1 is not needed). This approach is simple and can be described much more concisely.

Novelty is unclear, see above.

Reproducibility is somewhat unclear, since details about the proposed methods are missing (e.g., how many partitions are used). This point can be addressed if the authors make the implementation available, though.


**Strength And Weaknesses:**

Strengths:

S1. Can handle large graphs

S2. I/O mostly sequential and interleaved with communication

S3. Seems to provide reasonable results in study

Weaknesses:

W1. Novelty unclear; related work cited but otherwise ignored

W2. Simplistic & purely heuristic, no analysis

W3. Results far from SOTA

W4. Experimental study setup shaky

Details:

On W1. The most relevant related work to this work is MariusGNN (cited here as Marius++). There is no discussion of how this work relates to MariusGNN, both technically and empirically. This is not acceptable. In fact, MariusGNN seems to use a more refined approach and perform experiments on (partly) larger graphs, smaller machines, with faster speed ups. (I coincidentally found a very short paragraph on Marius++ in the appendix after writing this review. I consider this discussion insufficient.)

On W2. The method proposed here is rather simplistic; e.g., is does not consider to exchange loaded partitions individually, but only all at once. More generally, the entire approach is driven by heuristics, but there is no analysis that justifies the approach. There is also no discussion of what could go wrong.

On W3. The test accuracies reported in Tab. 2 seem to be far away from SOTA. E.g., for ogbn-papers100M, this paper gives a test accuracy of 0.65 with SAGE. The OGB leaderboard lists an accuracy of .78 with GraphSAGE and goes up to >.85 in all of the top-10 approaches.

On W4. The experimental setup uses machines with more memory than used by the proposed methods. For evaluating out-of-core methods, this is not good: this memory may be used as a disk cache by the OS, for example.


**Summary Of The Paper:**

Proposes an out-of-core training approach for message-passing GNNs on very large graphs. Key idea is to partition the nodes + their features on disk, load a few partitions into memory, and then build batches on the subgraph induced by the currently loaded nodes as well as the most frequent nodes, and repeat. Also reports on a small experimental study.


**Summary Of The Review:**

Potentially useful approach, but falls short on novelty, comparison to related work, analysis, and quality of results. I recommend to reject this paper.

---

> ### Author Response · Authors · 2022-11-19
> **Response to Reviewer p8K8**
>
> We would like to thank the reviewer for the valuable feedback. Before addressing any concern, we would like to first point out several inaccuracies/misunderstandings in the review and hopefully provide a better basis for the evaluation of our work.
>
> > The most relevant related work to this work is MariusGNN (cited here as Marius++). There is no discussion of how this work relates to MariusGNN, both technically and empirically.
>
> > I coincidentally found a very short paragraph on Marius++ in the appendix after writing this review. I consider this discussion insufficient.
>
> We actually do have discussions of Marius++ in the main text (Section 2, when discussing related works) and provide a pointer to the appendix later (Section 4.1, when explaining accuracy results). These discussions briefly cover the major limitations of Marius++. Since we did not have access to Marius++’s artifacts then, the empirical discussion is limited to a cross comparison of accuracy numbers presented in each paper. We notice that a revision of the Marius++ (thus renamed to MariusGNN) was uploaded to arxiv on Oct. 11, only 2 weeks before the submission of our paper, which leaves us little time to compare the work systematically. A detailed comparison between MariusGNN will be provided in this rebuttal and has been included in the paper revision.
>
> > W3. Results far from SOTA
>
> > The test accuracies reported in Tab. 2 seem to be far away from SOTA. E.g., for ogbn-papers100M, this paper gives a test accuracy of 0.65 with SAGE. The OGB leaderboard lists an accuracy of .78 with GraphSAGE and goes up to >.85 in all of the top-10 approaches.
>
> The SOTA results shown by the reviewer come from the ogbn-products dataset (https://ogb.stanford.edu/docs/leader_nodeprop/#ogbn-products), **not the mentioned ogbn-papers100M dataset**.
>
> Results for ogbn-products are also listed in Table 2 if the reviewer would like a comparison:  HB(-r) & HB reach 78.90% and 78.84% test F1 scores respectively, on par with the result from GraphSAGE+NS. The top-10 approaches in the leaderboard employ data augmentation or different GNN architectures and thus are incomparable with our results.
> The leaderboard of ogbn-papers100M (https://ogb.stanford.edu/docs/leader_nodeprop/#ogbn-papers100M) does not have results for vanilla GraphSAGE/GAT/GIN models, but our results are in line with previous works [1][2] that evaluated GraphSAGE on this dataset.
>
> > Reproducibility is somewhat unclear, since details about the proposed methods are missing (e.g., how many partitions are used).
>
> In our original submission we already list our hyper-parameter choices in Appendix C.3. A pointer to this appendix is provided in the main text (Section 4.1).
>
> ---
>
> [1] Global Neighbor Sampling for Mixed CPU-GPU Training on Giant Graphs.
>
> [2] Accelerating Training and Inference of Graph Neural Networks with Fast Sampling and Pipelining.

---

> > ### Author Response · Authors · 2022-11-19
> > **Response to Reviewer p8K8 (cont'd)**
> >
> > Now we address the reviewer’s remaining concerns.
> >
> > **W1. Comparison with related work MariusGNN.**
> >
> > We provide a detailed comparison with MariusGNN below. This discussion has been added into the updated paper in Appendix D.2.
> >
> > MariusGNN [3] is the only public work we are aware of that tackles out-of-core training of GNNs. Both MariusGNN and HierBatching work on pre-partitioned graph datasets and load parts of the graph in the memory for training. MariusGNN proposes two modes for training node-prediction GNNs:
> > * Sequential mode: when the training nodes and its induced subgraph could fit into the main memory, MariusGNN pins all training nodes and selects partitions randomly to form a partial in-memory graph. The graph remains unchanged throughout an epoch.
> > * Dispersed mode: when the sequential mode is not applicable, MariusGNN assumes the training nodes are dispersed across partitions. It then loads partitions into the in-memory buffer in a semi-random fashion until all training nodes appear once in the memory.
> >
> > Our approach is similar to the dispersed mode of MariusGNN, with several key differences: 1) we identify the importance of min-cut graph partitioning while MariusGNN simply uses random partitioning; 2) Marius in dispersed mode doesn’t have static cache; 3) MariusGNN doesn’t use mega-batch reuse to hide extra IO latency.
> >
> > An important component in MariusGNN is COMET, a fine-grained partition replacement policy. In addition to the number of partitions $p$ and in-memory partitions $c$, it specifies one extra tunable parameter $l\le c$ corresponding to the number of partitions to be replaced at each step. If put under MariusGNN’s terminology, HierBatching actually implements a COMET policy by making $l = c$. We argue that it is not a limitation of HierBatching. The value of $l$ actually has no influence on the overall IO traffic in the dispersed mode. Instead, a smaller $l$ would lead to less data randomness in the mega-batch and could affect the model accuracy. As a side note, MariusGNN suggests to set $l$ as large as possible for link prediction tasks (Section 4, no guidance on choosing $l$ is provided for node prediction tasks, though).
> >
> > Compared to our work, the main concern with MariusGNN is that it employs no techniques to make up for the lost neighbors during the mega-batch construction. This has not been an big issue because the datasets used in MariusGNN are “easier” in a sense: e.g. for ogbn-papers100M if we only keep edges connected by training nodes for training and discard the **remaining 99% edges**, the model could still achieve a test accuracy of 62.95%, an accuracy drop of merely 2% from training with the full graph. To reveal the risk of accuracy degradation in MariusGNN, we construct a new dataset by changing the data splits of ogbn-arxiv: 10% of nodes are chosen randomly as the training nodes and the rest as validation nodes. We compare the validation F1 scores of different approaches (we use the same set of hyper-parameters as in our paper):
> >
> > * NS (training with full graph): 69.61±.07
> > * HB (Nd / Np=1/16, 1% static cache): 69.64±.08
> > * MariusGNN (sequential mode, c/p=1/8): 67.08±.14
> > * MariusGNN (dispersed mode, c/p=1/8): 67.29±.10
> >
> > With this “harder” dataset, MariusGNN suffers from more significant accuracy losses, while HB still retains the model accuracy with NS.
> >
> > > In fact, MariusGNN seems to use a more refined approach and perform experiments on (partly) larger graphs, smaller machines, with faster speed ups.
> >
> > * We argue that HierBatching and MariusGNN are refined in different ways. For example, HierBatching focuses more on retaining the model accuracy and proposes techniques like static cache with proven optimal metrics. MariusGNN, on the other hand, pays more attention to the reduction of IO traffic by introducing the COMET policy.
> > * If we consider only node prediction tasks which are the target of our work, the largest dataset used in our paper (original Mag240M) is larger than MariusGNN (Mag240M-Cites).
> > * We will present HB’s results using less or the same amount of memory as MariusGNN when addressing W4.
> > * MariusGNN in the sequential mode is faster than our approach mainly because of less IO traffic, but we demonstrated this approach could lead to much worse model accuracies under certain situations. Besides, the better runtime performance is partially due to MariusGNN’s native (C++) implementation of training pipelines, while ours is mostly based on Python and DGL. The choice comes with a cost for MariusGNN, though. For example, the extensibility of MariusGNN is much poorer and it is difficult to reuse the rich GNN ecosystem the community has built with Python.
> >
> > [3] MariusGNN: Resource-Efficient Out-of-Core Training of Graph Neural Networks

---

> > > ### Author Response · Authors · 2022-11-19
> > > **Response to Reviewer p8K8 (cont'd)**
> > >
> > > **W2. The method proposed here is rather simplistic; e.g., is does not consider to exchange loaded partitions individually, but only all at once.**
> > >
> > > As has been discussed in the last response, exchanging loaded partitions individually would not bring any benefits to our approach but introduce one more hyper-parameter to tune. Thus, we do not think that it is a disadvantage of our approach.
> > >
> > > > More generally, the entire approach is driven by heuristics, but there is no analysis that justifies the approach. There is also no discussion of what could go wrong.
> > >
> > > We acknowledge that the training procedure in HierBatching deviates from the standard practice of SGD due to techniques like partitioning, static caching and mega-batch reuse. Our contribution, as a result, is largely empirical in nature. However, we argue that the empirical evidence provided in our paper is somewhat surprising: the GNN model converges as well as, if not better than the standard neighbor sampling approach in all the tested 15 experiment combinations (Table 2, Figure 3). Thus, we treat it more as an opportunity to motivate the theoretical analysis of non-standard GNN mini-batching and sampling techniques. We observe a similar trend in other deep learning areas: e.g. data echoing [4] is first proposed as an empirical technique to accelerate CNN/Transformer training, which motivates several works to rigorously analyze the convergence of non-standard batching schemes and beyond [5][6].
> > >
> > > **W3. Accuracy issues.**
> > >
> > > Already resolved.
> > >
> > > **W4. Inappropriate experimental setup.**
> > >
> > > In the original paper, we chose a unified 128GB memory budget for both DGL-mmap and HB just to simplify our complicated machine setups. HB could run with a lower memory budget. In the table below, we list the minimal amount of memory required by HB for the 3 large datasets. We also provide the training time per epoch under the given memory budgets. Note that the experiments are conducted with the local cluster and should compare with Table 6 in the paper. Since we are keeping improving the implementation of HB, the training time reported here is faster than the number in the original table.
> > >
> > > Dataset (SAGE) | ogbn-papers100M | Mag240M-C | Mag240M
> > > --- | --- | --- | ---
> > > Memory Budget | 48GB | 60GB |128GB
> > > Epoch Time (s) | 155.3 | 139.3 |1271.0
> > >
> > > Other concerns from the reviewer:
> > > > The exposition is clear, but unnecessarily wordy and overly formal (e.g., Alg 1 is not needed). This approach is simple and can be described much more concisely.**
> > >
> > > We simplified the algorithm description in Alg. 1 and Section 3.4 hoping that it would appear less verbose to the reviewer.
> > >
> > >
> > > [4] Faster Neural Network Training with Data Echoing
> > >
> > > [5] Stochastic Optimization with Laggard Data Pipelines
> > >
> > > [6] A General Analysis of Example-Selection for Stochastic Gradient Descent

---

### Official Review · Reviewer_Hx4Y · 2022-10-21

**Confidence:** 4
**Correctness:** 4
**Technical Novelty And Significance:** 2
**Empirical Novelty And Significance:** 2
**Recommendation:** 5

**Clarity, Quality, Novelty And Reproducibility:**

Clarity excellent
Quality moderate
Novelty low
Reproducibility excellent

**Strength And Weaknesses:**

Strengths

1. Training GNNs on giant data is important, and out-of-core training is interesting as it makes training more accessible for practitioners.
2. The paper is well-written with excellent clarity and logic flow.
3. The codes are open-source


Weakness
1. The novelties of the proposed techniques are limited. Partitioning a graph into strongly connected clusters is a standard method to achieve spatial locality in graph processing, DistDGL also uses it for the distributed training of GNNs. Caching hot nodes in has been used by [a, b] to handle limited GPU memory. Moreover, overlapping computation and IO is a common technique in system design.

2. HierBatching incurs accuracy loss, which may not be acceptable for some critical scenarios (e.g., recommendation, where marginal accuracy improvement translates into large revenue).

3. The experiments can be improved. Figure 4 shows that HierBatching uses over 50% of the time for disk IO, which makes me wonder whether training with HierBatching is cheaper than using distributed in-memory solutions such as DistDGL. Note that DistDGL already conducts locality-aware graph partitioning, which reduces cross-machine communication. By using less time for IO, DistDGL may be cheaper than HierBatching is we purchase instances from AWS. Another interesting experiment is to show how the accuracy and training time of HierBatching changes with the amount of main memory.

[a] https://arxiv.org/pdf/2111.05894.pdf
[b] https://dl.acm.org/do/10.5281/zenodo.6347456/full/


**Summary Of The Paper:**

This paper proposes HierBatching, a framework that conducts out-of-core training for GNNs on giant data on small main memory. HierBatching organizes the graph into closely connected partitions and loads some partitions to main memory to conduct training for locality. The nodes with the largest degrees are cached in main memory as they are accessed frequently. Pipelining is used to overlap disk read and computation. Experiment results show that HierBatching runs significantly faster than a naïve solution that directly reads disk.

**Summary Of The Review:**

I like that the paper tackles an important problem and conducts solid engineering. However, the utilized techniques are well-know and have limited novelty.

---

> ### Author Response · Authors · 2022-11-19
> **Response to Reviewer Hx4Y**
>
> We are happy to know that the reviewer agrees with the importance of the problem setting and finds our evaluation efforts solid. Below, we respond to the concerns expressed by the reviewer.
>
> **W1. The novelties of the proposed techniques are limited. Techniques commonly exist in other systems.**
>
> We believe the novelty of our paper is not sourced from the techniques themselves but the combinations of these techniques with the under-explored problem setting. They usually serve different purposes than the existing works. We address to the author’s comments on the three techniques one by one in the following:
>
> > Partitioning a graph into strongly connected clusters is a standard method to achieve spatial locality in graph processing, DistDGL also uses it for the distributed training of GNNs.
>
> The “spatial locality” mentioned by the reviewer here has a different meaning than what we presented in the paper.  We refer to spatial locality in terms of the physical layout of nodes’ data (e.g., features) on the storage, rather than the structural proximity of nodes in a partition. In fact, HierBatching works with any graph partitioning method (e.g. random partitioning) and could achieve spatial locality regardless, as long as partitions are laid out contiguously. As shown in the table below, with random partitioning our method has similar running time as Metis partitioning, which is not the case with DistDGL. In DistDGL, the quality of graph partitions (#edge cuts) is directly related to the communication overheads during training. Therefore, DistDGL would perform much worse with random partitioning.
>
> Table: ogbn-papers100M, SAGE, same set of hyper-parameter as Table 4
> Partitioner | Metis | Random
> --- | --- | ---
> Test F-1 (%).| 65.01±.11 | 64.45±.03
> Epoch Time (s) | 155.3 | 149.7
>
> The quality of graph partitioning affects the accuracy in our approach, though. The reason behind choosing Metis is discussed in the last paragraph of Sec 3.1.
>
> > Caching hot nodes has been used by [a, b] to handle limited GPU memory.
>
> Likewise, pinning hot nodes in the static cache is not designed to reduce the IO traffic (there is *no IO traffic* during mini-batch sampling in HierBatching), but to make up for the lost neighbors due to graph partitioning. The theoretical motivation is provided in Sec 3.2.
>
> > Moreover, overlapping computation and IO is a common technique in system design.
>
> We agree with the comment. However, in HierBatching, instead of applying the conventional pipelining technique alone, which may not be enough to hide IO latency, we effectively combine pipelining with mega-batch reuse to enhance our latency hiding capability.
>
> **W2. HierBatching incurs accuracy loss.**
>
> Table 2 shows that the accuracy loss is minor in most cases: we report less than **0.2%** accuracy loss with HB in **10 out of 15** experiments; the average accuracy loss is merely **0.042%**. Meanwhile, in the following table we provide the minimum amount of main memory required by different systems to train ogbn-papers100M. With HB, the main memory requirements are reduced by more than **4x** compared with in-memory DGL. We believe that under most realistic scenarios the accuracy drops are not significant enough to discount the benefits of our approach.
>
> Table: ogbn-papers100M, SAGE, same set of hyper-parameter as Table 4
> System | DGL (NS) | HB | DistDGL
> --- | --- | --- | ---
> Machine Setup | 1x 216GB | 1x 48GB | 4x 54GB
> Epoch Time (s) | 105.4 | 155.3 | 66.5
>
>
> **W3. Cost efficiency compared with DistDGL.**
>
> We run DistDGL with 4 nodes in our local cluster and record the running time in the table above. Each DistDGL node is regulated to use at most 54GB main memory so the aggregated memory is equal to the in-memory DGL system. We observe that DistDGL does not scale very well in our setup. If we use the metric of “#nodes$\times$ epoch-time” to compare the cost efficiency (regardless of memory difference), HB would be 1.71x better than DistDGL.

---

### Official Review · Reviewer_wEnk · 2022-10-23

**Confidence:** 3
**Clarity, Quality, Novelty And Reproducibility:** The paper is well written. The idea i…
**Correctness:** 3
**Technical Novelty And Significance:** 2
**Empirical Novelty And Significance:** 2
**Recommendation:** 5

**Strength And Weaknesses:**

## Strength:
The experiments show up to 20 times speedup than in-memory DGL on several large graphs. The analysis of hyper-parameters is in detail.

## Weakness:
1. My main concern is the convergence of the training because both static cache and reuse of mega-batch can change the sampling distribution of nodes.
2. I guess the performance including speedup and accuracy is also highly dependent on the parameters, such as mega-batch size and reuse times. Do you need to tune these parameters?

## Questions:
1. Table 5 provides the partitions (N_d) in mega-batch for each dataset, but I’m not sure how the authors decide it. I think there is a trade-off between accuracy and data loading time when using different N_d.
2. How many mega-batches are in the main memory? I assume the number is two so that the main memory can read the next mega-batch from external memory and construct minibatches at the same time.

**Summary Of The Paper:**

In order to train GNN efficiently on large graphs with a single machine and external storage, this paper proposes a locality-aware training scheme to reduce the data movement time. First, it partitions the graph using METIS. The main idea is to save a mega-batch, which is randomly sampled from partitions in the main memory, and construct mini-batches from the mega-batch. To alleviate the performance from insufficient neighbors in mega-batch, it permanently saves nodes with the highest degree. Reusing each mega-batch several times and pipelining further reduces the total training time.

**Summary Of The Review:**

Reducing the data movement of GNN training on CPU-GPU nodes has been extensively studied recently. The paper proposes a straightforward technique.  However, it is unclear whether the technique can be applied to more general GNN tasks.

---

> ### Author Response · Authors · 2022-11-19
> **Response to Reviewer wEnk**
>
> Thank you for the valuable feedback. We now address the reviewer's concerns below.
>
> **W1. My main concern is the convergence of the training because both static cache and reuse of mega-batch can change the sampling distribution of nodes.**
>
> We are aware that the training in HierBatching deviates from the standard practice of SGD due to techniques like partition-based batching, static caching and mega-batch reuse. However, we argue that the empirical evidence provided in our paper is somewhat surprising: the GNN model converges as well as, if not better than the standard neighbor sampling approach for all the 15 combinations (Table 2, Figure 3). Thus, we treat it more as an opportunity to motivate the theoretical analysis of non-standard GNN mini-batching and sampling techniques. We observe a similar trend in related deep learning areas: e.g. data echoing [1] is first proposed as an empirical technique to accelerate CNN/Transformer training, which motivates several works to rigorously analyze the convergence of non-standard batching schemes and beyond [2][3].
>
> **W2.1 Sensitivity of speedup and accuracy w.r.t. mega-batch size and reuse times.**
>
> The speedup and the model accuracy is not highly dependent on the mega-batch size $N_d$ or the reuse ratio p.
>
> Generally, a smaller $N_d$ has a negative impact on the accuracy because of more lost neighbors in a mega-batch, but empirically we find that the accuracy is insensitive to a wide range of $N_d$ (see table below) The speedup is also relatively insensitive: our algorithm requires loading each partition once into the memory, regardless of $N_d$. The overall IO traffic, as a result, remains constant. When IO dominates the training time, the speedup would not change drastically.
>
> Table: ogbn-papers100M, SAGE, same hyper-parameters as the paper except $N_d$
>
> $N_d$ | 256 | 1024 | 4096 | in-memory
> --- | --- | --- | --- | ---
> Test F-1 (%) | 64.57±.32 | 65.01±.11 | 64.99±.22 | 65.06±.18
> Epoch Time (s) | 182.9 | 155.3 | 161.3 | 105.4
>
> As for reuse raio p, we already demonstrate in the paper that mega-batch reuse causes no harm to the final accuracy (columns HB vs. HB(+r) of Table 2), but instead improves the convergence speed of the training (Figure 3).
>
> **W2.2 Do you need to tune these parameters (e.g. $N_d, p$)?**
>
> **Q1.1 How to choose $N_d$.**
>
> We'd like to point the reviewer to our response to W3 [here](https://openreview.net/forum?id=WWD_2DKUqdJ&noteId=wU41o7mpBx) since they are similar questions.
>
> **Q1. Trade-off between accuracy and data loading time when using different N_d.**
>
> Should be resolved in our response to W2.1.
>
> **Q2. Number of mega-batches in memory.**
>
> We use double buffering to overlap mega-batch construction and mini-batch training, so at most 2 mega-batches will be in the memory simultaneously.
>
> **S1. Reducing the data movement of GNN training on CPU-GPU nodes has been extensively studied recently. The paper proposes a straightforward technique.**
>
> We share the observation with the reviewer that GNN training on CPU-GPU nodes has been a popular research topic. However, we'd like to highlight that our paper takes one more step and targets an under-explored scenario where the main memory is insufficient to hold all the training data. The scenario might seem similar to the case of mini-batch GNN training motivated by insufficient GPU memory, but it is not, as explained below:
>
> * The slowdown of random access to the external storage is much more severe than the main memory due to hardware limitations. Simply reusing data tiering or caching [4][5][6] won’t work well, because 1) they can not eliminate random accesses to the storage, and 2) the benefits of caching are highly dataset/parameter dependent (since their cache hit rates vary a lot across datasets/parameters). Our solution does not suffer from such problems.
> * An out-of-core training system does not have convenient access to the complete dataset (including graph structures) due to memory constraints, while most existing single-node solutions assume they do. This challenge motivates us to the hierarchical batching scheme. However, care must be taken with the mega-batch construction: over-sparsified mega-batches could hurt the accuracy badly. We solve the problem with the usage of static cache and min-cut graph partitioning. Although straightforward, these techniques prove to be very effective and make implementation easier.
>
> [1] Faster Neural Network Training with Data Echoing
>
> [2] Stochastic Optimization with Laggard Data Pipelines
>
> [3] A General Analysis of Example-Selection for Stochastic Gradient Descent
>
> [4] Global Neighbor Sampling for Mixed CPU-GPU Training on Giant Graphs.
>
> [5] Graph Neural Network Training with Data Tiering.
>
> [6] GNNLab: A Factored System for Sample-based GNN Training over GPUs.

---

> > ### Author Response · Authors · 2022-11-19
> > **Response to Reviewer wEnk (cont'd)**
> >
> > **S2. It is unclear whether the technique can be applied to more general GNN tasks.**
> >
> > We could extend our system to link prediction as follows. Rather than making a mini-batch of nodes to compute the loss, we make a mini-batch of edges to compute the loss. We can keep our mega-batch formalism, but rather than making $N_p/N_d$ mega-batches to form one training epoch, we’ll need to come up with a policy and create a certain number of mega-batches to form one training epoch. For this policy, imagine that we have a $N_p\times N_p$ matrix; each time we will extract an $N_d\times N_d$ principal submatrix to form one mega-batch. After a certain number of times, each location of the $N_p\times N_p$ matrix is used at least once, thus completing one training epoch.

---

### Official Review · Reviewer_bDQL · 2022-10-26

**Confidence:** 2
**Correctness:** 4
**Technical Novelty And Significance:** 2
**Empirical Novelty And Significance:** 3
**Recommendation:** 6

**Clarity, Quality, Novelty And Reproducibility:**

The writing is reasonably clear and provides a clear distinction regarding the problem, motivation, and novelty of the proposed solution. The experiments represent reasonable evidence to support the claims made by the authors. I believe the results could be reproduced based on the information provided in the text.

**Strength And Weaknesses:**

Strengths
-------------------
- The experiments support the author's motivation to provide a static cache for a small number of highly connected nodes and re-using nodes within a mega-batch while loading additional nodes from disk during training.
- Ablation studies provide extensive data regarding the tradeoffs between the accuracy and performance of the different approaches.
- As the graphs of interest for GNN training continue to increase in size it will be of imperative importance to make effective use of the memory hierarchy of the entire system. As demonstrated in Section 4.2 the authors are able to train extremely large graphs using a fraction of the DRAM memory and incurring a moderate amount of degradation in performance.

Weaknesses
-------------------
- The major weakness is the lack of theoretical motivation to support the training scenario the authors describe in section 3.3. However, the authors acknowledge this weakness openly in the paper and pose it as an interesting area for future work.
- Though the ablation studies were beneficial to support the author's claims regarding the use of a static cache to store the nodes with the largest degrees it may have been more straightforward to use a typical metric for cache effectiveness, such as hit-rate per minibatch.
- The proposed scheme introduces yet another set of hyperparameters that much be tuned per input graph to yield the best results in terms of performance and accuracy, as illustrated in Figure 5.
- The costs and impact of different graph partitioning strategies are not provided (ie Metis partitions may be of low quality depending on a number of factors, would that have any notable impact on the accuracy of the results?).

**Summary Of The Paper:**

In this paper, the authors describe a hierarchical strategy for storing large batches of GNN training data in multiple levels of memory in a single system. By partitioning the storage of GNN training data between the extremely large out-of-core disk, smaller but faster CPU DRAM main memory, and much more limited but fastest GPU DRAM memory they are able to train GNNs on graphs that are far too large to fit into either GPU or CPU DRAM memory much faster than naive strategies. To achieve this the authors compare and contrast the training accuracy vs performance tradeoff of their approach vs the traditional neighbor sampling strategy used by default in DGL. A detailed strategy for detailing the high variability in the degree of nodes in a given graph is also given and the authors propose a decomposition of the cache into dynamic and static components to keep the highest utilized nodes throughout the training process in fast memory. An ablation study shows the utility of their approach and the impact on both training accuracy and speed.

**Summary Of The Review:**

Overall the paper addresses an important issue regarding the training of large graphs on machines with limited memory. The authors present a strategy to mitigate the limitations associated with training large GNNs and yield a training procedure that effectively utilizes the hierarchy of multiple memory systems of different capacities and performance characteristics. Though the authors mention the need to minimize the graph cut metric in Section 3.1, was Metis the easiest choice or the best? Also, in Section 3.1 the authors mention an out-of-core partitioner was used when the graph exceeded the memory capacity, please add a reference. It may be worth considering a simpler way to motivate the utility of the static cache size in terms of the hit rate over an epoch or similar metric of typical interest for studying the utility of caches. More information regarding the graph partitioning hyperparameters is provided in the Appendix, Section C.3, were the number of partitions chosen based only on memory considerations? It would be useful if the comment in section 3.1 regarding partitions that are too large or small could be supported by data.

---

> ### Author Response · Authors · 2022-11-19
> **Response to Reviewer bDQL**
>
> We appreciate the constructive feedback from the reviewer. In the following we address the reviewer's concern one by one.
>
> **W1. Lack of theoretical motivation to support the mega-batch reuse.**
>
> We acknowledge that the training example selection in HierBatching deviates from the standard practice of SGD. Our contribution, as a result, is largely empirical in nature. However, we argue that the empirical evidence provided in our paper is somewhat surprising: with mega-batch reuse the GNN model converges as well as, if not better than the no-reuse cases in all the tested 15 experiment combinations (Table 2, Figure 3). Thus, we treat it more as an opportunity to motivate the theoretical analysis of non-standard GNN mini-batching and sampling techniques. We observe a similar trend in other deep learning areas: e.g. data echoing [1] is first proposed as an empirical technique to accelerate CNN/Transformer training, which motivates several works to rigorously analyze the convergence of non-standard batching schemes and beyond [2][3].
>
> [1] Faster Neural Network Training with Data Echoing
>
> [2] Stochastic Optimization with Laggard Data Pipelines
>
> [3] A General Analysis of Example-Selection for Stochastic Gradient Descent
>
> **W2. A better metric to evaluate the effectiveness of static cache.**
>
> Common metrics used for evaluating caches such as hit rates are not a good fit in our use cases, because we use the caches differently: the dynamic cache and static cache are used to construct a mega-batch; next, the mini-batches are sampled *within* the mega-batch, so there will be *no cache misses* at all (i.e. 100% cache hits, either serviced by the dynamic cache or the static cache).
>
> As described in Sec. 3.2, our objective for the optimal static cache is based on the maximization of degree increases for nodes in the mega-batch, Therefore, we validate the impacts of the static cache on the node degrees using the ogbn-arxiv dataset in Figure 2. We think that the evidence provided in Figure 2 together with the accuracy improvements in Table 2 is sufficient to support our usage of the static cache.
>
> **W3. Hyper-parameter tuning.**
>
> Let $r_d = N_d/N_p$. Our approach introduces 4 extra tunable hyper-parameters ($N_p, r_d, r_s, p$), but fortunately most of them ($N_p, r_d+r_s, p$) could be either **pre-determined** based on the hardware constraints *before training* or easily chosen *after running several mega-batches*. In our original submission we already provide the guidance for choosing these hyper-parameters in Appendix B. The split between $r_d$ and $r_s$ is the only “knob” that requires tuning during the training. Both Figure 5 and our tuning experience suggest that the final accuracy under different $r_s$ follows a U-shaped curve and thus is easy to tune. In practice, we also suggest $[0.001, 0.01]$ as the starting interval for tuning $r_s$ since it generally works very well for us (e.g. we choose $r_s=0.01$ for all datasets).
>
> *Choosing $p$*. We describe here how to decide $p$ in a mechanical way, since it is not included in the original submission. Assume the time to load and construct and mega-batch is $T_{load}$ and the time without reuse is $T_{comp}$. Since the two stages are pipelined, the latency between the processing of two mega-batches is max{$T_{load}$ , $T_{comp}$}. When $T_{comp} < T_{load}$, we use a reuse ratio of $p \approx T_{comp}/T_{load}$ to hide the remaining data-loading latency but guarantee that $T_{comp}$ never becomes the bottleneck. In terms of accuracy, we already demonstrate in the paper that mega-batch reuse causes no harm to the final accuracy (see columns HB vs. HB(+r) of Table 2 ), but instead improves the convergence speed of the training (Figure 3).
>
> **W4. Choice of graph partitioners.**
>
> Metis is chosen in our paper mainly due to two reasons:
> * Relatively good partitioning quality. We would like to lose as few neighbors as possible during partitioning, so min-cut graph partitioners are desirable. Metis is one of the best-known partitioners that produce high-quality partitions and run efficiently.
> * Ease of use. The GNN framework we build HierBatching on, DGL, provides a convenient API for calling Metis on its graph data structure.
>
> We actually find that Metis is reliable in delivering good results and versatile in functions. Apart from Metis, we provide preliminary results with random partitioning below.
>
> Dataset      | ogbn-arxiv, SAGE | ogbn-products, SAGE | ogbn-papers100M, SAGE
> --- | --- | --- | ---
> Partitioner      | Metis / Random  | Metis / Random    | Metis  / Random
> Test F-1 (%)   | 71.42$\pm$.26 / 70.17$\pm$.39 | 78.84$\pm$.37 / 78.18$\pm$.31 | 65.01$\pm$.11 / 64.45$\pm$.03
>
> Random partitioning usually incurs significant amounts of edge cuts and fails our objective of reducing lost neighbors. As shown in the table, HB with random partitioning always yields lower model accuracies than Metis.

---

> > ### Author Response · Authors · 2022-11-19
> > **Response to Reviewer bDQL (cont'd)**
> >
> > We also respond to the questions raised in the summary but not yet addressed above:
> >
> > > In Section 3.1 the authors mention an out-of-core partitioner was used when the graph exceeded the memory capacity, please add a reference.
> >
> > We added the reference in the paper revision.
> >
> > > It would be useful if the comment in section 3.1 regarding partitions that are too large or small could be supported by data.
> >
> > The parameter $N_p$ determines the partition sizes. We conduct a sensitivity study with the ogbn-papers100M dataset and SAGE by adjusting the values of $N_p$. The results are summarized below:
> >
> > ogbn-papers100M, SAGE, $r_d$=1/16, $r_s$=1%
> > $N_p, N_d$ | $2^{10}, 2^6$ | $2^{12}, 2^8$ | $2^{14}, 2^{10}$ | $2^{20}, 2^{16}$
> > --- | --- | --- | --- | ---
> > Test F-1 (%) | 64.68±.27 | 64.99±.32 | 65.01±.11 | N/A
> > Epoch Time (s) | 167.1 | 162.9 | 155.3 | 1545.1
> >
> > (The experiments were run on a machine of the local cluster, so the running time should be compared with numbers in Table 6. Besides, since we keep optimizing the implementation of HierBatching after the original submission, the training time reported here is generally faster than Table 6.)
> >
> > As the partition size becomes larger (i.e. smaller $N_p$), we eventually observe a benign decrease of the test accuracy (~0.3%), but within a wide range of choices for $N_p$ (e.g. 4096 ~ 16384) the model accuracy is not affected. The training time remains little changed for $N_p$  <= 16384 since the size of a single partition is large enough (>3MB) and the time spent on disk IO is constant. These observations suggest the choice of $N_p$ is easy in practice. In the other extreme, as we push the number of partitions ($N_p=2^{20}$) the average partition size becomes smaller than ideal (roughly tens of KBs). As a result, the training slows down by 10 times, wasting most of the time on IO. With this configuration, the training can’t finish in reasonable time so the accuracy is not reported.

---

### Decision · Program_Chairs · 2023-01-20

**Decision:**

Reject

**Justification For Why Not Higher Score:**

NA

**Justification For Why Not Lower Score:**

NA

**Metareview: Summary, Strengths And Weaknesses:**

The paper proposes a memory hierarchy aware batching scheme to efficiently train large graph neural network on modest compute resources which may not have very large main memory. The paper proposed a variety of useful heuristics and provide good experimental comparisons to demonstrate speedups.

However, in the context of existing works the ideas presented are a combination of existing known ideas and follows from several recent lines of work. For example using graph reordering so that memory accesses are contiguous can significantly enhance graph traversal minimizing cache misses  was discussed in recent NeurIPS 2022 paper  "Graph Reordering for Cache-Efficient Near Neighbor Search".








**Summary Of Ac-Reviewer Meeting:**

In light of existing idea, the main contribution of the paper is a system development. However, from a systems perspective the evaluation is not rigorous. Mere accuracy and overall speedup does not qualify for a rigorous system paper evaluations.  Reviews also echo similar lack of excitement in their reviews and rebuttal did not bring back the excitement.

A system paper needs scaling experiments (how does increasing memory affect caching performance etc.), attribution of speedup to caching behavior experiments, etc.  The evaluation should itself provide more insights into different aspects of the system behavior.